# Transcriptome Analysis Deciphers *Trichoderma koningiopsis* C5-9 Strategies against Plant Pathogen *Botrytis cinerea*

**Min Yuan** [1,†], **Chunliu Zuo** [1,†], **Wen Xu** [1,†], **Li Zhang** [2], **Xinyue Guo** [1], **Xinyue Yan** [1], **Songyang Li** [1], **Yanling Li** [1], **Lan Zhang** [1], **Jiaqi Geng** [3] and **Yali Huang** [3,*]

1   College of Life Sciences, North China University of Science and Technology, Tangshan 063210, China; yuanmin308@163.com (M.Y.); cell2020@163.com (C.Z.); xuwbio@ncst.edu.cn (W.X.); gxy18232288302@163.com (X.G.); yanxinyue1114@163.com (X.Y.); m13041338595@163.com (S.L.); lingling609727@163.com (Y.L.); zhanglan1374@sohu.com (L.Z.)
2   College of Bioscience and Engineering, Hebei University of Economics and Business, Shijiazhuang 050061, China; zhangli-amy@126.com
3   College of Environmental Sciences and Engineering, Hebei University of Science and Technology, Shijiazhuang 050018, China; gengjiaqi0811@163.com
*   Correspondence: huangyali2291@163.com
†   These authors contributed equally to this work.

**Abstract:** The plant pathogen *Botrytis cinerea* (*B. cinerea*) causes severe plant diseases worldwide. *Trichoderma* is widely used as a biocontrol agent against *B. cinerea* through multiple biocontrol mechanisms. However, *Trichoderma* spp. with high biocontrol efficiency against *B. cinerea* under low-temperature conditions are barely reported. This study aimed to find potential low-temperature resistance biocontrol *Trichoderma* spp. against *B. cinerea*, and to characterize the biological principles underlying the activity of *Trichoderma*. *Trichoderma koningiopsis* (*T. koningiopsis*) C5-9 could fully overgrow a *B. cinerea* colony at 16 °C in a dual-culture assay. Treatment of cucumber leaves with *T. koningiopsis* C5-9 fermentation broth using the dipping method prior to *B. cinerea* inoculation significantly reduced the necrotic lesion diameter, with an inhibition rate of 55.30%. *T. koningiopsis* C5-9 could be successfully cultivated using the mycelia of *B. cinerea* as a carbon source at 16 °C. Transcriptomic analysis indicated that the origin recognition complex, organic substance catabolic process, and peroxisome were involved in the responses of *T. koningiopsis* C5-9 to *B. cinerea*. The findings of this study not only identified *T. koningiopsis* C5-9 as a potential biological control agent inhibiting *B. cinerea* under low-temperature conditions, but also provided new insights to develop a deeper understanding of the activity of *Trichoderma* against *B. cinerea* for plant protection.

**Keywords:** *Trichoderma koningiopsis* C5-9; *Botrytis cinerea*; biological control; plant protection





## 1. Introduction

Mycoparasitism can be dated to about 400 million years ago and involves a fungus (prey or host) being parasitized by another fungus (mycoparasite or predator) and serving as a nutrient source in a mycoparasitic interaction. This activity is significant when the prey is a plant pathogen because it can provide a valuable strategy for the biological control of plant pests in agriculture (biocontrol) [1–5]. Hypocrea/*Trichoderma*, the most studied biocontrol agent in agriculture, could compete with plant pathogens for space and nutrients to kill fungal plant pathogens [6–9]. In addition, some *Trichoderma* biocontrol strains could interact with plant roots, promote plant growth, and prime a plant's systemic resistance to plant pathogens, such as *Trichoderma harzianum* T22 and *Trichoderma longibrachiatum* H9 [10–13]. These traits enable *Trichoderma* strains to act as avirulent plant symbionts and thus decrease the agricultural industry's heavy reliance on chemical pesticides and fertilizers in crop production.

Consequently, some studies have demonstrated that cell wall hydrolytic enzymes and a variety of antagonistic secondary metabolites are involved in the recognition of

a predatory fungus and in mycoparasitic responses [7,14,15]. However, the process of mycoparasitism appears to be complicated. Except for a mechanism partially shared by several *Trichoderma* spp., some strain-specific mechanisms are employed by different *Trichoderma* spp. in the following steps, including detecting the host fungi, priming the gene expression, and producing hydrolytic enzymes and secondary metabolites. Therefore, different *Trichoderma* spp. may have variable biocontrol performance in their interactions with distinct prey [8,16,17].

The successful utilization of *Trichoderma* for agricultural applications requires an enhanced understanding of the molecular mechanisms underlying the activity of *Trichoderma* in mycoparasitic interactions.

Cucumbers, one of the most economically important vegetables, are very susceptible to infections caused by the plant pathogenic fungus *Botrytis cinerea* (*B. cinerea*) under low-temperature conditions [18–20]. Currently, *Trichoderma* spp. with high biocontrol activity against *B. cinerea* under low-temperature conditions are rarely reported. Finding potential low-temperature-resistant biocontrol *Trichoderma* spp. against *B. cinerea* is thus warranted. In this study, *Trichoderma koningiopsis* (*T. koningiopsis*) C5-9 worked directly and effectively against *B. cinerea* under low-temperature conditions. Furthermore, the RNA-seq approach was used to investigate the gene expression changes of the mycoparasite *T. koningiopsis* C5-9 during its physical contact with the pathogen *B. cinerea*. The results allow for a more comprehensive interpretation of the mycoparasitic process and will lay the foundation for selecting genetic traits to improve *Trichoderma* biocontrol strains for plant protection.

## 2. Materials and Methods

### 2.1. Microorganism

Eight *Trichoderma* strains with an antagonistic capability against *B. cinerea* were obtained from Banbi Hill in Chengde, Hebei Province, China and stored at North China University of Science and Technology. The *B. cinerea* stain (B05.10) was provided by Dr. Jihong Xing at Hebei Agricultural University in China.

### 2.2. Dual-Culture Assay

The in vitro antagonistic activity against *B. cinerea* by *Trichoderma* strains was determined through a dual-culture assay. Five-millimeter mycelial disks of both *B. cinerea* and different *Trichoderma* strains were placed on either side of potato dextrose agar (PDA) plates 10 mm away from the edge of a 90 mm Petri dish. The plates inoculated with *B. cinerea* alone served as the control. Three biological replications were performed for each *Trichoderma* strain. All the plates were incubated at 16 °C, and observations were made on a daily basis. The radial growth of *B. cinerea* mycelia was photographed and measured using ImageJ software (version: Fiji). The image data of a maximum of four weeks of observations were used for the measurement of *B. cinerea* growth, and the inhibition rate by *Trichoderma* was determined using the following formula [21–23]:

$$I\% = (R1 - R2)/R1 \times 100,$$

where I is the inhibition rate and R1 (cm) and R2 (cm) are the radial growth of *B. cinerea* in the presence and absence of *Trichoderma* strains, respectively.

### 2.3. Molecular Identification of the C5-9 Strain

The molecular identification of the *Trichoderma* strain C5-9 was determined through the sequencing of the internal transcribed spacer (ITS) and translation elongation actor 1-alpha (tef-1α) gene. The genomic DNA of C5-9 was extracted using a Fungi Genomic DNA Extraction Kit (Cat. No. D2300, Solarbio Co., Beijing, China). The ITS and TEF-1α fragments of C5-9 were amplified, sequenced, and deposited in the GenBank database with accession numbers ON514163 and ON552552, respectively. The primers are described in Table S1. The PCR procedure was as follows: 95 °C for 30 s; followed by 40 cycles at 95 °C for 5 s, 60 °C for 10 s, and 72 °C for 30 s; 72 °C for 5 min; hold at 16 °C. Phylogenetic trees

were built using MEGA software (version: 11.0) according to ITS and TEF-1α sequences of C5-9 and other known *Trichoderma* strains in the GenBank database.

### 2.4. The Mycelial Growth Inhibition Experiment

Conidia were harvested by flooding the surface of *T. koningiopsis* C5-9 cultures with sterile water containing 0.1% Triton-X100. A 1 mL conidial suspension with a concentration of $1 \times 10^6$–$1 \times 10^8$ cfu/mL (colony-forming units per milliliter) was aliquoted and inoculated into 100 mL potato dextrose (PD) broth in a shaker at 16 °C and 180 rpm for 96 h. The fermentation broth was collected after removing the mycelium and freeze-dried to 1.5 g powder. The powder was resolved in water and filtered through a 0.22 μm microporous membrane to obtain a sterile fermentation broth with a concentration of 0.3 g/mL. In the mycelial growth inhibition experiment, PDA plates were prepared containing 7.5 mg/mL fermentation broth, and sterile water as the mock solution was added to the control group. A 5 μL conidial suspension of *B. cinerea* with a concentration of $1 \times 10^6$ cfu/mL was inoculated and placed in the center of each PDA plate. The plates were kept at 16 °C for four days, and the radial growth of *B. cinerea* mycelia was measured and recorded every 24 h. The inhibition rate of a *Trichoderma* fermentation broth was determined using the formula above. R1 (cm) and R2 (cm) are the radial growth of *B. cinerea* in the PDA plates containing a *T. koningiopsis* C5-9 fermentation broth or mock solution, respectively.

### 2.5. Efficacy of T. koningiopsis C5-9 Fermentation Controlling B. cinerea in Cucumber

The freeze-dried powder of *T. koningiopsis* C5-9 fermentation broth was prepared as described above and resolved to a 7.5 mg/mL C5-9 fermentation solution. The third leaves of three-leaf-stage cucumber plants were dipped in the C5-9 fermentation solution and air-dried in an airflow cabinet. Cucumber leaves dipped in mock solution (water) served as the negative control. Then, 5 mm mycelial disks of *B. cinerea* were placed on the cucumber leaves, which were incubated in 1/2 Ms (Murashige and Skoog) growth medium in a moist box for three days. Finally, the lesion development of *B. cinerea* infection in all treated cucumber leaves was measured, and inhibition rates were calculated using the formula above. Three cucumber leaves (three replicates) were examined for each treatment. R1 (cm) and R2 (cm) are the radial growth of pathogen *B. cinerea* in the cucumber leaves, which were dipped with *T. koningiopsis* C5-9 fermentation solution or water, respectively, prior to *B. cinerea* infection.

### 2.6. Preparation of Deactivated B. cinerea Mycelia and Cultivation of T. koningiopsis C5-9

A conidial suspension of *B. cinerea* was aliquoted into a PDB medium with a final concentration of $1 \times 10^4$–$1 \times 10^6$ cfu/mL and further grown at 28 °C on a rotary shaker for 48 h. The mycelia of *B. cinerea* were collected by centrifugation, filtered through a cheesecloth, washed three times with sterile water, and autoclaved for 15 min at 121 °C for deactivation. Then, 0.5% (*w/v*) deactivated mycelia of *B. cinerea* (Bc) or 0.5% (*w/v*) glucose (G) was added as a carbon source to the minimal medium to make two different media, which were named Bc and G medium, respectively. The ingredients of the minimal medium were as follows: 0.14% (*w/v*) $(NH_4)_2SO_4$; 0.69% (*w/v*) $NaH_2PO_4$; 0.2% (*w/v*) $KH_2PO_4$; 0.1% (*w/v*) peptone; 0.03% (*w/v*) $MgSO_4.7H_2O$; and 0.03% (*w/v*) urea; the pH was 5.0 [7,14].

For the cultivation of *T. koningiopsis* C5-9, a 3 mL conidial suspension with a concentration of $1 \times 10^6$–$1 \times 10^8$ cfu/mL was aliquoted and inoculated into 100 mL Bc or G medium for a further 24 h incubation in a shaker at 16 °C and 180 rpm. The mycelia of *T. koningiopsis* C5-9 were collected by filtering through a cheesecloth, washed three times with sterile water, and frozen in liquid nitrogen for RNA sequencing. Since the deactivated mycelia of *B. cinerea* did not cause interference in the RNA-seq, they were not removed when collecting the mycelia of *T. koningiopsis* C5-9. Each process was performed with three biological replicates.

## 2.7. Glucose Quantification

Glucose quantification was performed through the 3,5-dinitrosalicylic acid (DNS) method, and DNS reagents were prepared as previously described [24,25]. First, 185 g of sodium potassium tartrate were added to 500 mL distilled water and heated until dissolved. Next, 6.3 g of DNS and 262 mL of 2 M NaOH were added to the above solution. The volume was then adjusted to 1.0 L with distilled water. Standard solutions contained 0, 0.02, 0.04, 0.06, or 0.08 mg D (+)-glucose monohydrate, respectively, in 1 mL distilled water, and then 1 mL of DNS solution was added into each reaction to reach a final volume of 2 mL. The reaction was conducted at 100 °C for 15 min, followed by cooling to room temperature, and absorbencies were measured at 540 nm. The standard curve of glucose was created based on the contents of glucose in standard samples and the corresponding absorbencies. For measuring the content of glucose in the cultures, 0.5 mL solution from the G group was added to 0.5 mL distilled water to a final volume of 1 mL.

## 2.8. Transcriptome Analysis by RNA-Seq

RNA-seq was performed by [8] double-ended sequencing on an Illumina HiSeq™2000 platform. The reference genome for alignment was acquired from https://www.ncbi.nlm.nih.gov/genome/56074 (accessed on 26 September 2021). The raw sequencing data were uploaded into Genome Sequence Archive (GSA) under the accession number CRA006882. Raw sequences were processed to remove low-quality reads and adaptor sequences. The NCBI nonredundant protein (Nr), Swiss-Prot, Pfam, STRING, Gene Ontology (GO), Kyoto Encyclopedia of Genes and Genomes (KEGG) databases were employed for gene annotation. FPKM (Fragments Per Kilobase of exon model per Million mapped reads) was calculated to determine the gene expression levels, and differentially expressed genes (DEGs) were determined by $|\log_2 \text{FoldChange}| \geq 1.00$ and $p$-value $< 0.05$. Principal component analysis (PCA) and GO/KEGG enrichment analysis of (DEGs) were performed as described previously [26–28].

## 2.9. Quantitative Reverse Transcriptase-PCR (qRT-PCR) Analysis

The total RNA of *T. koningiopsis* C5-9 from each sample was extracted using the RNA extraction kit (Cat. No. LS1040, Promega Co., Madison, WI, USA). First-strand cDNAs were synthesized using a RevertAid kit (Cat. No. K1622, Thermo Fisher Scientific, Waltham, MA, USA). qRT-PCR was performed on the ABI CFX Connect Real-Time PCR machine (ABI 7500) using an SYBR green reagent (Cat. No. FP205, Tiangen Biotech Co., Ltd., Beijing, China). The primers used in this study are described in Table S1. Actin gene was used as the reference [29].

## 2.10. Statistical Analysis

The results regarding fungal *B. cinerea* inhibition, necrotic lesion development, and gene expression in the qRT-PCR assay were subjected to a one-way analysis of variance (ANOVA). Statistical significance of differences among different groups was determined by Student's *t*-test [30,31].

## 3. Results

### 3.1. A Trichoderma Strain, C5-9, Was Isolated by Inhibiting B. cinerea under Low-Temperature Conditions

In order to isolate *Trichoderma* strains antagonistic to pathogenic fungus *B. cinerea* under low-temperature conditions, a total of eight *Trichoderma* strains were isolated and analyzed by in vitro dual-culture assay at 16 °C. All eight isolates showed inhibition activity against *B. cinerea*, and the strain C5-9 exhibited the highest antifungal activity among the eight isolates. The *Trichoderma* strain C5-9 grew faster and stopped the growth of *B. cinerea* after contact. A clear inhibition edge and barrage zone appeared after six days' growth and was subsequently overgrown by *Trichoderma* C5-9. The radial growth of pathogen *B. cinerea* varied from 7.06 ± 0.01 cm to 3.06 ± 0.03 cm in the presence of *Trichoderma* C5-9, and the

inhibition rate was 59.7% (Figure 1 and Table S2). The data from other *Trichoderma* strains were not shown.

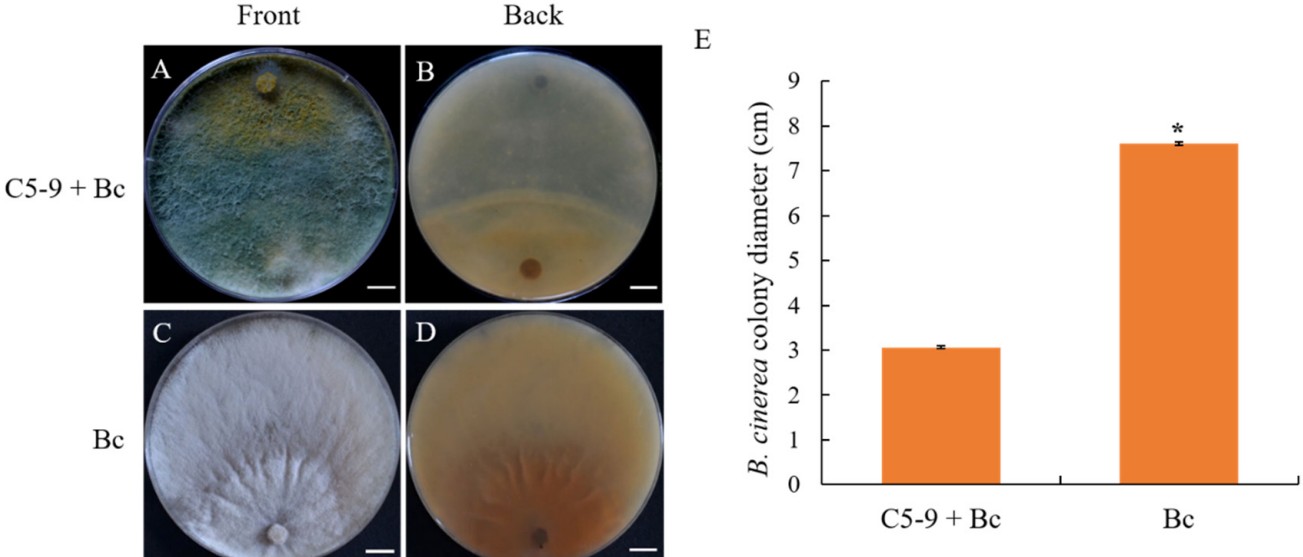

**Figure 1.** *Trichoderma* C5-9 significantly inhibited the mycelial growth of the plant pathogen *Botrytis cinerea* (*B. cinerea*) in the dual-culture assay. Dual cultures of C5-9 and *B. cinerea* were co-incubated on PDA plates, and C5-9 grew faster and fully overgrew the *B. cinerea* colony (**A,B**). The culture, containing only *B. cinerea*, was inoculated in parallel as a control group (**C,D**). Growth was documented when a clear inhibition zone was observed at the back of the plates (**E**). Means ± SD (*n* = 3) labeled with * are significantly different (*p* < 0.05) compared with the control according to a Student's *t*-test.

### 3.2. Morphological and Molecular Identification of Trichoderma C5-9

The morphological characteristics of the strain C5-9 were first examined to determine its species. The colony radius was 15–16 mm after four days and covered the entire plate after six days on PDA under 16 °C, at which time yellow–green conidia started to form in the center of the colony at high rates. Aerial mycelium was abundant. No distinctive odor or diffusing pigment was produced. The conidiospore and conidia of *Trichoderma* C5-9 were observed under a microscope. The main conidiophores were long and straight, and the secondary branches were alternated, occurred at acute angles, and had a slight curvature to the main branch. The secondary base branches were longer than the distal branches. The conidia were elliptical and smooth, with a 2–6 μm diameter (Figure 2A–C). The *Trichoderma* C5-9 strain exhibited the typical morphological features of *T. koningiopsis*.

Phylogenetic analysis based on DNA sequence was further performed to confirm the identity of this strain. Based on the ITS sequence and TEF-1α gene, this *Trichoderma* strain shared the highest homology and clustered together with *T. koningiopsis* strains (Figure 2D,E). The C5-9 strain was accordingly named *T. koningiopsis* C5-9.

### 3.3. Antifungal Activity of the Fermentation Broth from the T. koningiopsis C5-9

To further investigate the antifungal activity of the products from *T. koningiopsis* C5-9, the mycelial growth of plant pathogenetic fungi *B. cinerea* was monitored in a PDA solid medium containing *T. koningiopsis* C5-9 fermentation broth or mock solution at 16 °C. The *B. cinerea* colony diameter ranged from 0.61 ± 0.01 cm to 1.71 ± 0.02 cm, from 1.87 ± 0.02 to 4.12 ± 0.02, and from 3.07 ± 0.02 to 6.36 ± 0.03 after 2, 3, and 4 days, respectively, of growing in a medium supplemented with *T. koningiopsis* C5-9 fermentation broth or mock. *T. koningiopsis* C5-9 fermentation broth significantly inhibited the mycelial growth of pathogen *B. cinerea*, with the highest inhibition percentage being 64.30% (Figure 3 and Table S3). To determine whether *T. koningiopsis* C5-9 could control *B. cinerea* infection in cucumbers, the leaves from four-leaf-stage cucumber plants were dipped with *T. konin*-

*giopsis* C5-9 fermentation broth or mock solution prior to *B. cinerea* inoculation, and the resistance to *B. cinerea* infection was subsequently assessed by measuring the necrotic lesion diameter. On the third day after inoculation of *B. cinerea*, the diameters of necrotic spots were $1.23 \pm 0.09$ cm and $2.75 \pm 0.13$ cm in the cucumber leaves, which were pre-dipped with *T. koningiopsis* C5-9 fermentation broth and mock solution, respectively. Treatment of cucumber leaves with *T. koningiopsis* C5-9 fermentation using the dipping method prior to *B. cinerea* inoculation significantly reduced the necrotic lesion diameter with an inhibition rate of 55.30% (Figure 4 and Table S4).

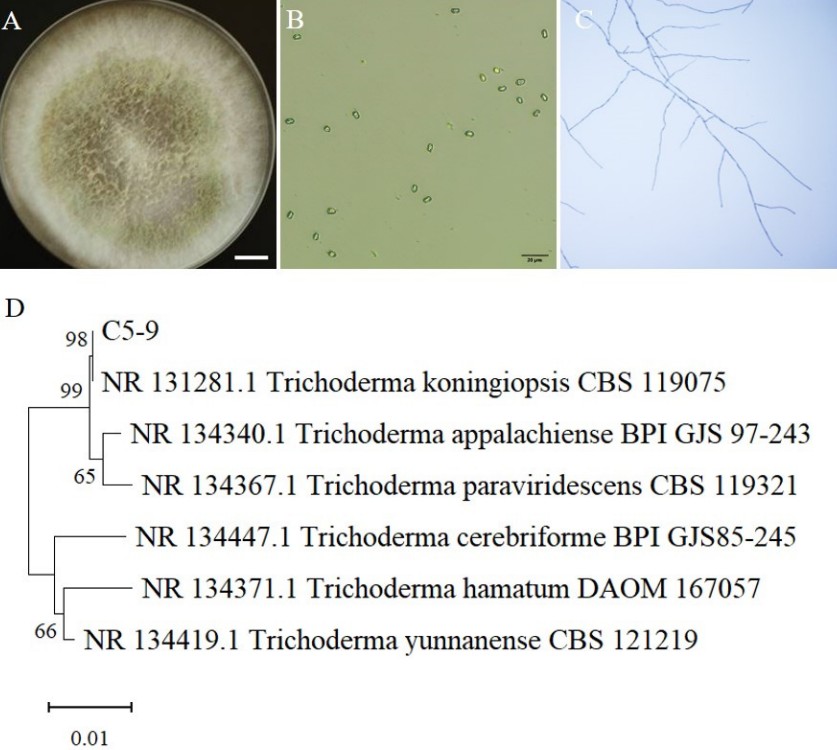

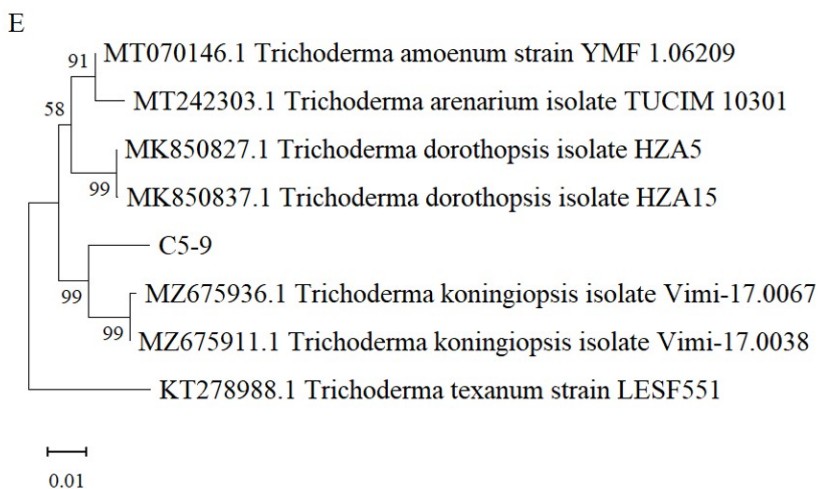

**Figure 2.** Morphological and molecular characterization of *Trichoderma* C5-9. (**A**) Cultures on PDA medium at 16 °C after one week. Scale bar: 1 cm. (**B**) The morphology of conidia. Scale bar: 20 μm. (**C**) The morphology of mycelia. Scale bar: 1 mm. (**D,E**) Neighbor-joining phylogenetic trees were built based on ITS and TEF-1α sequences of the *Trichoderma* species.

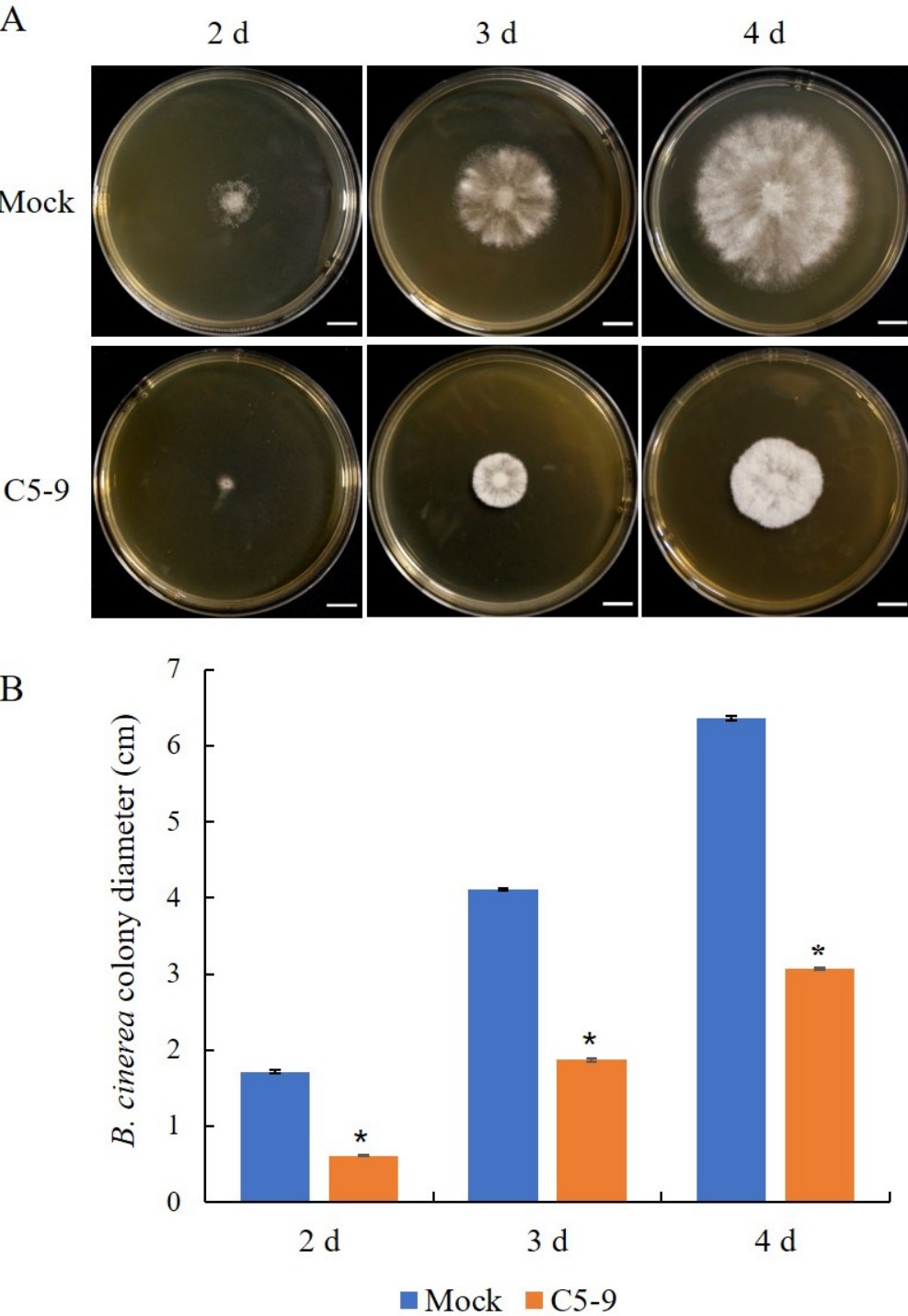

**Figure 3.** Effect of the fermentation broth produced by *Trichoderma koningiopsis* (*T. koningiopsis*) C5-9 on the mycelial growth of *Botrytis cinerea* (*B. cinerea*). (**A**) The mycelial growth of *B. cinerea* was monitored on a daily basis (2, 3, and 4 days) in the PDA solid medium containing the *T. koningiopsis* C5-9 fermentation broth or mock control under 16 °C. (**B**) The *B. cinerea* colony diameter was measured, and the fermentation broth produced by *Trichoderma* C5-9 significantly inhibited *B. cinerea* growth in the C5-9 group. Means ± SD (*n* = 3) labeled with * are significantly different (*p* < 0.05) compared with the control according toa Student's *t*-test.

Taken together, the biocontrol assays indicated that *T. koningiopsis* C5-9 could directly parasitize *B. cinerea*.

### 3.4. T. koningiopsis C5-9 Could Utilize B. cinerea as a Carbon Source

Mycoparasitism is an ancient lifestyle in which one fungus (prey or host) is parasitized by another fungus (mycoparasite or predator) through serving as a nutrient source in a mycoparasitic interaction. To further explore the mycoparasitic ability of *T. koningiopsis* C5-9 on *B. cinerea*, *T. koningiopsis* C5-9 were cultivated in two different liquid cultures, which were supplemented with the mycelia of *B. cinerea* (the Bc group) or glucose (the G group), respectively, as a carbon source. The mycelia of *T. koningiopsis* C5-9 became visible after 24 h incubation at 16 °C in both media, indicating that *T. koningiopsis* C5-9 was able to successfully grow using the mycelia of *B. cinerea* or glucose as a carbon source (Figure 5A). Meanwhile, a visible surplus of the deactivated mycelia of *B. cinerea* was observed in the Bc group, and the content of glucose decreased to 0.15% from 0.5% in the G group (Figures 5B and S1). The results not only revealed that the carbon sources were not exhausted and the biomass of *T. koningiopsis* C5-9 was increasing in both media, but also lay the foundation for us to monitor the gene expression of *T. koningiopsis* C5-9 cultivated with two different carbon sources. Finally, 103.87 $\pm$ 1.39 mg and 235.03 $\pm$ 2.25 mg mycelia were harvested for each replicate in the G and Bc group, respectively (Figure 5C and Table S5). Heavier mycelia were collected in the Bc group, because the rest of the deactivated mycelia of *B. cinerea* was not removed.

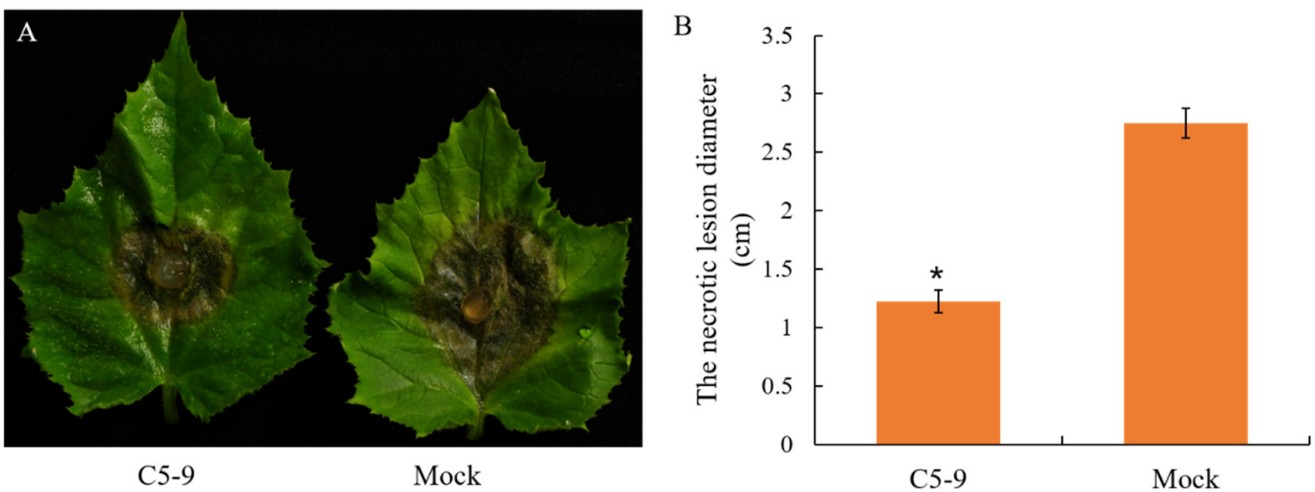

**Figure 4.** Efficacy of the fermentation broth produced by *Trichoderma koningiopsis* (*T. koningiopsis*) C5-9 on suppression of infection by *Botrytis cinerea* (*B. cinerea*) on cucumber leaves. (**A**) The infection of *B. cinerea* on cucumber leaves, which were pre-dipped in the C5-9 fermentation broth or mock (water) solution, respectively. (**B**) Lesion sizes were determined by measuring the necrotic lesion diameter three days after *B. cinerea* inoculation. Means $\pm$ SD ($n = 3$) labeled with * are significantly different ($p < 0.05$) compared with the control according toa Student's *t*-test.

### 3.5. Transcriptome Analysis Deciphering the Strategy of T. koningiopsis C5-9 Mycoparasitism against B. cinerea

Since *T. koningiopsis* C5-9 could grow using the mycelia of *B. cinerea* as a carbon source, RNA-seq analysis was performed to examine the biological principles underlying the *Trichoderma* actions. *T. koningiopsis* C5-9 was cultivated in two different liquid cultures, supplemented with the mycelia of *B. cinerea* (Bc) or glucose (G) as a carbon source. The mycelia of *T. koningiopsis* C5-9 were collected 24 h post-inoculation for RNA-seq to monitor the changes in gene expression. A total of 329,598,454 bp raw reads were generated, and 328,941,224 bp clean reads were obtained with GC content over 53.67% after quality control. The reads were mapped onto the *T. koningiopsis* (ascomycetes) genome, and the percentage of uniquely mapped ones exceeded 87.87%. For gene annotation, the most annotated genes (8999) were retrieved from the Nr database, followed by the Pfam database (5898). A

total of 4156 and 3279 genes were assigned to the GO and KEGG databases, respectively (Tables S6–S8).

In order to facilitate the comparison between samples and identify sample clusters with high similarity, Principal Component Analysis (PCoA) was performed to reduce the dimensionality of the large amount of gene expression information contained in the six samples to a few unrelated principal components. Significant separation among treatments was observed with PC1 and PC2 explaining 45.52% and 20.97%, respectively, of the overall variance (Figure 6A). To explore the genes associated with the *T. koningiopsis* C5-9 response to *B. cinerea*, DEGs were screened in the *T. koningiopsis* C5-9 under two different culture conditions. A total of 1431 DEGs with 672 upregulated genes and 759 downregulated genes were identified in *T. koningiopsis* C5-9, which were cultivated in a minimal medium supplemented with the mycelia of *B. cinerea* or glucose as a carbon source (Figure 6B).

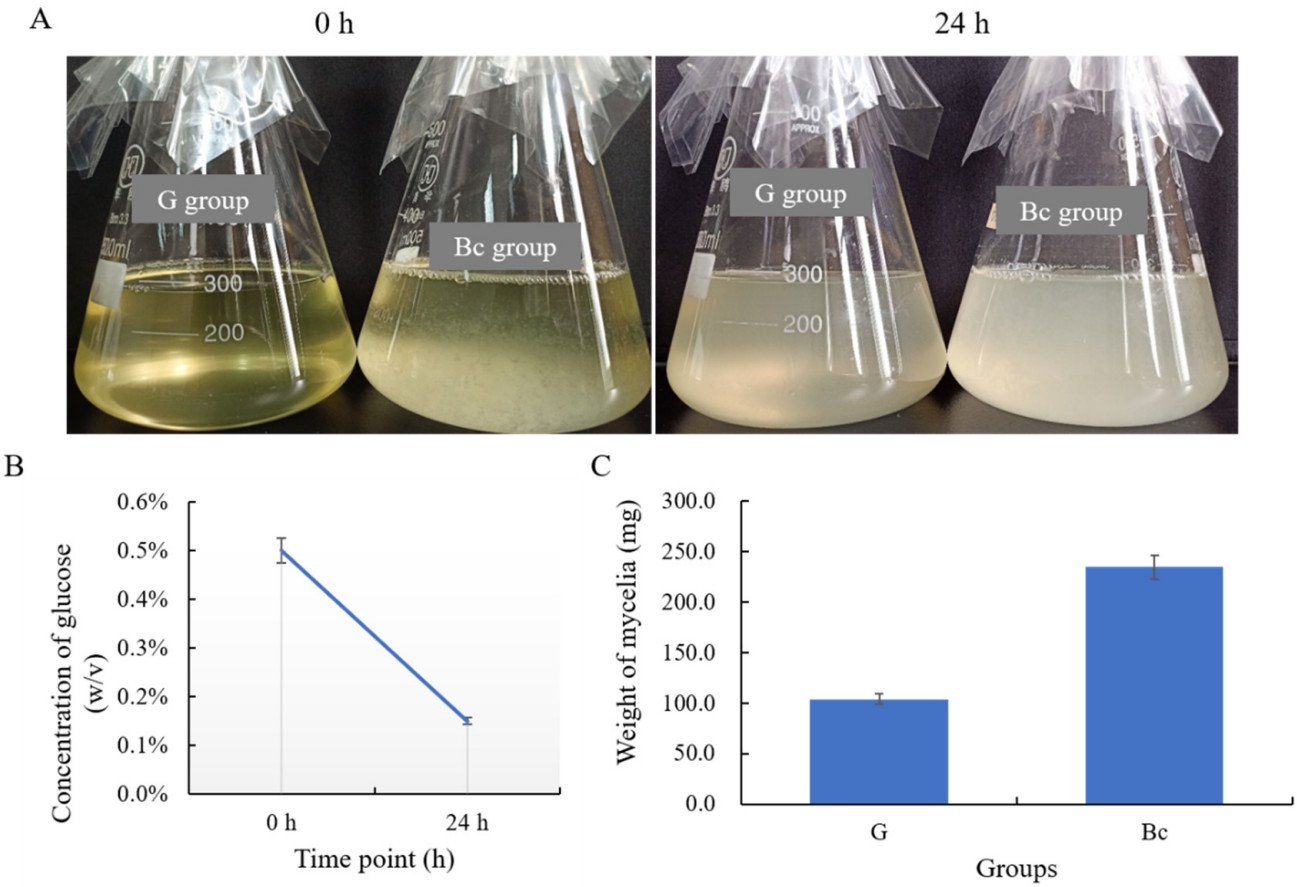

**Figure 5.** *Trichoderma koningiopsis* (*T. koningiopsis*) C5-9 was able to grow using the mycelia of *B. cinerea* or glucose as a carbon source. (**A**) The mycelium of *T. koningiopsis* C5-9 became visible after 24 h incubation under 16 °C in the medium supplemented with the mycelia of *B. cinerea* (Bc) or glucose (G) as a carbon source. (**B**) The content of glucose was estimated in the G group using the 3,5-dinitrosalicylic acid (DNS) method. The content of glucose decreased to 0.15% from 0.5% after 24 h incubation. (**C**) The mycelia were harvested and measured in the two groups.

Functional significance was further examined through GO and KEGG enrichment analyses according to the upregulated and downregulated DEGs. A total of 177, 304, and 157 GO terms were significantly enriched based on total, upregulated, and downregulated DEGs, respectively (Tables S9–S11). More GO terms were enriched associated with upregulated DEGs, indicating that the upregulation of gene expression played essential roles in the *Trichoderma* actions. Furthermore, the top 10 enriched GO terms were explored based on the numbers of upregulated DEGs in the biological_process, molecular_function, and cellular_component categories, respectively. A variety of metabolic and catabolic processes

exhibited a significant difference in the biological_process category. Metabolic processes were mainly involved in the carbohydrate metabolic process, small molecule metabolic process, and lipid metabolic process. Catabolic processes specifically referred to the organic substance catabolic process. Correspondingly, organic acid transmembrane transporter activity was significantly enriched in the molecular_function category. Notably, the "peroxisome," "microbody," and "origin recognition complex" were found to be significantly enriched GO terms in the cellular_component category (Figure 7 and Table S12). Moreover, KEGG enrichment analysis was performed to identify specific pathways based on DEGs. A total of 14, 9, and 11 enriched KEGG terms were explored based on total, upregulated, and downregulated DEGs, respectively. Interestingly, the "biosynthesis of antibiotics" and "peroxisome" were found in the nine enriched KEGG terms according to the upregulated DEGs (Tables S13–S15).

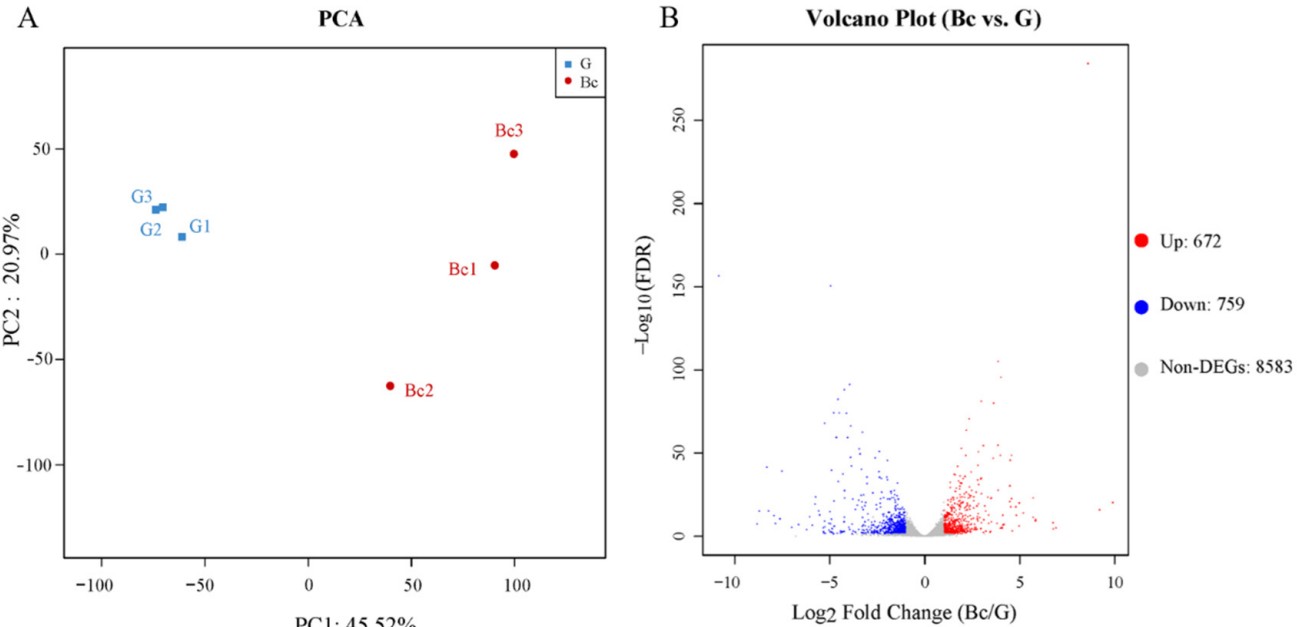

**Figure 6.** The gene expression of *Trichoderma koningiopsis* (*T. koningiopsis*) C5-9 was noticeably different when cultivated in the minimal medium supplemented with the mycelia of *B. cinerea* (Bc) or glucose (G) as a carbon source. (**A**) Significant separation was detected between two different groups. (**B**) Up- and downregulated DEGs shown as a scatter plot.

### 3.6. Confirmation of Gene Expression by qRT-PCR

To further confirm those processes were associated with the response of *T. koningiopsis* C5-9 to *B. cinerea*, we performed qRT-PCR to validate the expression of some DEGs involved in the origin recognition complex, organic substance catabolic process, and peroxisome, including MSTRG.11196 (peptidase), MSTRG.12273 (protease), MSTRG.14312 (E3 ubiquitin ligase), MSTRG.175 (glycosyl hydrolase), MSTRG.3184 (beta-1,3-glucanase), MSTRG.5600 (acetyltransferase), MSTRG.6748 (amino acid permease), and MSTRG.7917 (fungal specific transcription factor). The expression of those DEGs was significantly enhanced in *T. koningiopsis* C5-9 when cultivated with the mycelia of *B. cinerea* as a carbon source in the qRT-PCR results, which was consistent with the results of the RNA-seq analysis, indicating the reliability of the transcriptome analysis. The findings revealed that the origin recognition complex, organic substance catabolic process, and peroxisome played essential roles in the *T. koningiopsis* C5-9 responses to *B. cinerea* (Figure 8).

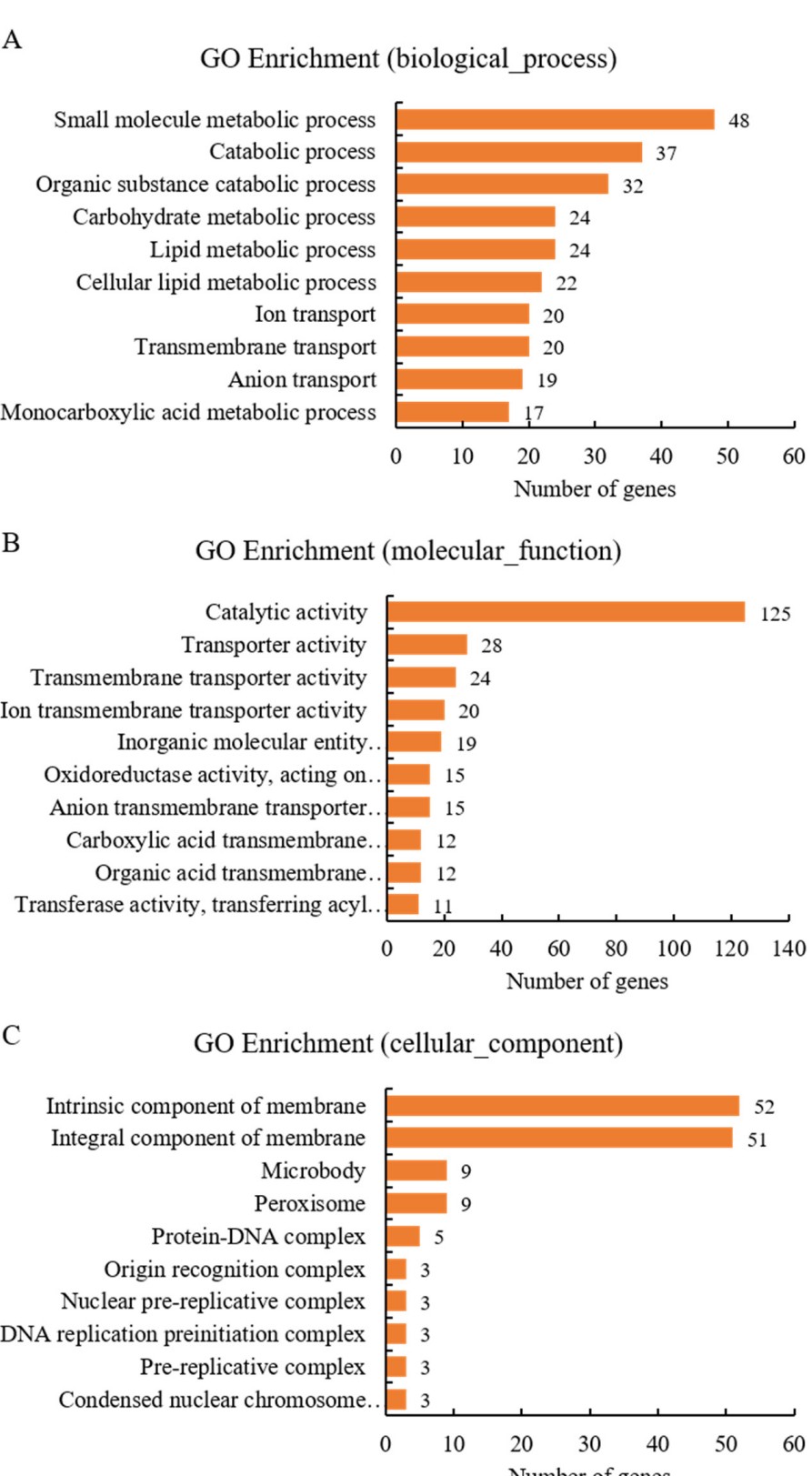

**Figure 7.** GO enrichment analysis based on corresponding upregulated DEGs in *Trichoderma koningiopsis* (*T. koningiopsis*) C5-9 cultivated in the media supplemented with the mycelia of *B. cinerea* (Bc) or glucose (G) as a carbon source, respectively. Top 10 GO terms among biological_process (**A**), molecular_function (**B**) and cellular_component (**C**) categories were shown based on upregulated genes ($p < 0.05$).

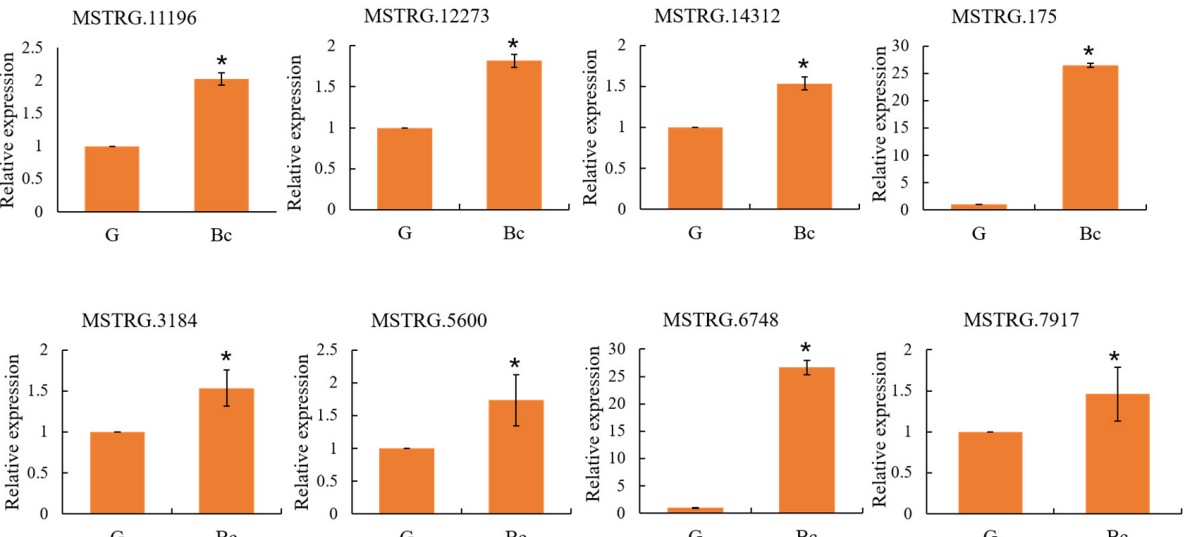

**Figure 8.** Validation of the expression of eight selected DEGs by RT-qPCR. Error bars indicate the mean ± standard deviation (SD). * $p < 0.05$, as determined by a Student's *t*-test.

## 4. Discussion

Grey mold, caused by the plant-pathogenic fungus *B. cinerea*, leads to a significant reduction in both the quantity and quality of crop production worldwide. Chemical fungicides controlling this pathogen pose a serious risk to human and environmental health. A *Trichoderma*-based biocontrol method has become a preferable alternative to control *B. cinerea*. Cucumbers, economically essential vegetables, are very susceptible to *B. cinerea* infection in low-temperature conditions [32–35]. Finding potential low-temperature resistance biocontrol *Trichoderma* spp. against *B. cinerea* is thus warranted.

### 4.1. T. koningiopsis C5-9 Is a Potential Biological Control Agent Inhibiting B. cinerea under Low-Temperature Conditions

*B. cinerea* causes grey mold in over 200 plant species, including a large number of crops, and is difficult to control because of its survival ability under unfavorable conditions, such as low temperature. However, low-temperature resistance biocontrol *Trichoderma* strains against *B. cinerea* are few in number at present [36–40]. In this study, *Trichoderma* C5-9 was demonstrated to directly suppress the pathogen *B. cinerea* growth under low-temperature conditions in a dual-culture assay. The fermentation broth produced by this strain effectively inhibited the mycelial growth of *B. cinerea* and protected cucumbers from *B. cinerea* infection. Furthermore, *T. koningiopsis* C5-9 could be successfully cultivated using the mycelia of *B. cinerea* as a carbon source. The results indicated the great potential of *T. koningiopsis* C5-9 to be utilized as a biological control agent (BCA) inhibiting *B. cinerea* infection under low-temperature conditions.

### 4.2. Mycoparasitism Is Strain-Specific as a Conserved Biocontrol Mechanism in Trichoderma

Understanding the biological principles underlying *Trichoderma* actions is critical for the science-based improvement of biocontrol agents to support agricultural applications. It was reported that mycoparasitism, competition, and antibiosis were involved in the direct interactions between *Trichoderma* spp. and *B. cinerea*. *Trichoderma* can directly antagonize plant pathogens by competing for space and nutrients, secreting antibiotics, and finally overcoming the growth of other fungal pathogens. During this process, *Trichoderma* spp. relied heavily on mycoparasitism to recognize, antagonize, and kill other fungi [6,41–43]. Cell wall-degrading enzymes (CWDEs), such as glucanase, chitinases, chitosanases, proteases, and glycosyl hydrolases, were demonstrated to be produced to degrade prey. For example, *Trichoderma harzianum* ALL 42 produced different hydrolases, including chitinases, α-1,3-glucanases, β-1,3-glucanases, proteases, and glucoamylases, when cultivated in a

minimal medium supplemented with cell walls from the phytopathogen *Fusarium solani* [7]. The β-1,3-glucanase and chitinase were produced by *T. koningiopsis* PSU3-2 to control the postharvest anthracnose in chili peppers [23]. Similarly, when *T. harzianum* ETS 323 was cultured with deactivated *B. cinerea* mycelia, significantly higher activities of the CWDEs were detected in media, including chitinases, proteases, β-1,6-glucanases, β-1,3-glucanases, and xylanases [14]. Moreover, several genomic studies have revealed that the genes encoding a variety of CWDEs were regulated in *Trichoderma* spp. when antagonizing prey [8,44,45].

Our findings in this study indicated that *T. koningiopsis* C5-9 could successfully grow using the mycelia of *B. cinerea* as a carbon source. RNA-seq and RT-qPCR assays showed that the genes encoding proteases and glucanases were upregulated in *T. koningiopsis* C5-9, when cultivated in the minimal medium supplemented with deactivated *B. cinerea* mycelia, which implied that cell wall degrading is a crucial step of this process. The results are in agreement with previous publications and also implied a mycoparasitic mechanism of *T. koningiopsis* C5-9 involved in biocontrol activity. However, no genes encoding chitinases were identified in this study. Moreover, various DEGs in this study were not identified in the transcriptome analysis using different *Trichoderma* strains and were not previously reported in mycoparasitism-related studies. This points to alternative strategies of mycoparasitism employed by different *Trichoderma* strains to attack different hosts. Similar clues were also identified from three different *Trichoderma* strains to the presence of alien hyphae. *T. virens*, *T. atroviride*, and *T. reesei* exhibited different transcriptomic responses in producing different secondary metabolites, proteases, gliotoxin, or cellulases [8].

Moreover, some DEGs did not have any match in the six selected databases, and their roles in mycoparasitism are still unknown, indicating that a distinct set of genes were regulated during *T. koningiopsis* C5-9 contact with the fungal pathogen *B. cinerea*. Meanwhile, the lack of annotated information in the current databases is a rising concern in the era of "big data," and further functional studies are necessary to validate the prediction data from large-scale analyses.

### 4.3. Early Recognition of the Host Is Critical for Successful Trichoderma Actions

Mycoparasitism is reported to be a multistep process in which physical contact among microorganisms is initiated by early recognition [36,46,47]. Through analyzing the DEGs in this study, the "origin recognition complex" and "peroxisome" GO terms were significantly enriched in the cellular_component category, with the corresponding genes significantly upregulated. The enrichment of the "origin recognition complex" indicated that early recognition of the host is critical for successful parasitism. Additionally, essential for coping with environmental stress by the *T. koningiopsis* C5-9 is the enrichment of "peroxisome" GO terms. Similarly, an L-amino acid oxidase (LAAO) was demonstrated to be secreted by *T. harzianum* ETS 323 when cultivated with deactivated *B. cinerea* mycelia as the carbon source [14]. $H_2O_2$ would be produced during the oxidization of L-amino acids by LAAO. We speculate that $H_2O_2$, together with other reactive oxygen species, played important roles during the *Trichoderma* parasitism process.

The mechanism underlying *Trichoderma* actions against *B. cinerea* appears to be complex, with both shared and specific traits belonging to different *Trichoderma* strains, which may be involved in the recognition of host fungi, the activation of intracellular regulatory gene networks, and the production of hydrolytic enzymes and secondary metabolites. Notably, there are reasons to believe that *T. koningiopsis* C5-9 also antagonizes *B. cinerea* through complex and combined strategies. Competing for space and nutrients (competition), production of CWDEs (mycoparasitism), and production of antibiotics (antibiosis) may be the main factors contributing to its success in antagonizing *B. cinerea*. Firstly, *Trichoderma* C5-9 directly suppressed the pathogen *B. cinerea* in a dual-culture assay under low-temperature conditions. Secondly, *T. koningiopsis* C5-9 could be successfully cultivated using the mycelia of *B. cinerea* as a carbon source, and the genes encoding a variety of CWDEs were up-regulated. Finally, the fermentation broth produced by *T. koningiopsis* C5-9 effectively inhibited the mycelial growth of *B. cinerea* and protected cucumbers from

*B. cinerea* infection. The KEGG terms associated with "biosynthesis of antibiotics" were significantly enriched, with correspondingly upregulated DEGs.

Genetic studies are required to validate the assigned functions for the genes identified by the large-scale analysis, which would be the next step for us to better understand the mechanism underlying *Trichoderma* actions. The genome-wide expression data in this study provide an opportunity to develop a deeper understanding of the *Trichoderma* parasitism process against the plant pathogen *B. cinerea*.

## 5. Conclusions

*T. koningiopsis* C5-9 grew faster, stopped the growth of *B. cinerea* after contact, and fully overgrew the *B. cinerea* colony at 16 °C in a dual-culture assay. A clear inhibition edge appeared with an inhibition rate of 59.7%. The mycelial growth of *B. cinerea* was significantly inhibited in a PDA medium supplemented with *T. koningiopsis* C5-9 fermentation broth, with an inhibition percentage of 64.30% at day 2. Treatment of cucumber leaves with *T. koningiopsis* C5-9 fermentation broth prior to *B. cinerea* inoculation significantly reduced the necrotic lesion diameter with an inhibition rate of 55.30%. Moreover, *T. koningiopsis* C5-9 was able to grow in a minimal medium supplemented with the mycelia of *B. cinerea* as a carbon source. Transcriptomic analysis and qRT-PCR assay demonstrated that the origin recognition complex, organic substance catabolic process, and peroxisome played essential roles in the *T. koningiopsis* C5-9 responses to *B. cinerea* with corresponding DEGs. The results indicated that *T. koningiopsis* C5-9 is a potential low-temperature resistance biocontrol *Trichoderma* sp. against *B. cinerea*, and the DEGs represent important resources to improve antagonistic activity against pathogenic fungi. The top 10 enriched GO terms in each category based on the numbers of upregulated DEGs will be a matter of future study on fungal pathogen control.

**Supplementary Materials:** The following supporting information can be downloaded at: https://www.mdpi.com/article/10.3390/microbiolres14030067/s1, Figure S1: Standard curves of glucose; Figure S2: Amplification of the ITS and TEF-1α fragments of C5-9; Table S1: List of primers used in this study; Table S2: Antifungal activity of *Trichoderma* C5-9 against *Botrytis cinerea* (*B. cinerea*) in a dual-culture assay; Table S3: Effect of the fermentation broth from *Trichoderma koningiopsis* (*T. koningiopsis*) C5-9 on the mycelial growth of plant pathogen *Botrytis cinerea* (*B. cinerea*); Table S4: Efficacy of the fermentation broth produced by *Trichoderma koningiopsis* (*T. koningiopsis*) C5-9 on suppression of *Botrytis cinerea* (*B. cinerea*) on cucumber leaves; Table S5: The mycelia were harvested and measured in the two groups; Table S6: Summary of the sequencing and assembly; Table S7: Mapped results of the RNA sequencing data; Table S8: Annotation result in different databases; Table S9: GO enrichment analysis based on DEGs of *Trichoderma koningiopsis* (*T. koningiopsis*) C5-9 cultivated in two different media; Table S10: GO enrichment analysis based on upregulated DEGs of *Trichoderma koningiopsis* (*T. koningiopsis*) C5-9 cultivated in two different media; Table S11: GO enrichment analysis based on downregulated DEGs of *Trichoderma koningiopsis* (*T. koningiopsis*) C5-9 cultivated in two different media; Table S12: Top 10 enriched GO terms in each category based on the numbers of upregulated DEGs; Table S13: KEGG enrichment analysis based on DEGs in *Trichoderma koningiopsis* (*T. koningiopsis*) C5-9 cultivated in two different media; Table S14: KEGG enrichment analysis based on upregulated DEGs in *Trichoderma koningiopsis* (*T. koningiopsis*) C5-9 cultivated in two different media; Table S15: KEGG enrichment analysis based on downregulated DEGs in *Trichoderma koningiopsis* (*T. koningiopsis*) C5-9 cultivated in two different media.

**Author Contributions:** Methodology, M.Y., W.X. and L.Z. (Li Zhang); validation, M.Y., W.X. and C.Z.; formal analysis, C.Z., X.G., X.Y., and J.G.; investigation, S.L., Y.L. and L.Z. (Li Zhang); data curation, L.Z. (Lan Zhang) and X.G.; visualization, X.Y. and L.Z. (Lan Zhang); writing—original draft preparation, M.Y.; writing—review and editing, Y.H.; supervision, M.Y. and Y.H.; project administration, M.Y. and Y.H. All authors have read and agreed to the published version of the manuscript.

**Funding:** This research was funded by the Key Research and Development Program of Hebei Province (20326513D), the Natural Science Foundation of Hebei Province (C2020209020), and the National Key Research and Development Program of China (2021YFD1901004).

**Data Availability Statement:** The raw data of RNA sequencing in this study was deposited into the Genome Sequence Archive (GSA) under the accession number CRA006882. The sequencing data of ITS and TEF-1α fragments of *T. koningiopsis* C5-9 were deposited in GenBank with the accession numbers ON514163 and ON552552, respectively.

**Conflicts of Interest:** The authors declare no conflict of interest.

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
