# Peer review of "Transcriptome Analysis Deciphers Trichoderma koningiopsis C5-9 Strategies against Plant Pathogen Botrytis cinerea"

_2036-7481, doi:10.3390/microbiolres14030067_

Round 1

Reviewer 1 Report

Title: Transcriptome Analysis Deciphers Trichoderma koningiopsis 2 C5-9 strategies Against Plant Pathogen Botrytis cinerea

The current manuscript has a good idea but needs some modifieds

Abstract:
-it is good, but the authors should consider the proposed changes for improving the clarity of the content.

Keyword: Good

-Introduction part is appropriate but a few things are needed for further improvements, especially the study hypothesis should be added.

Add some studies about the study with highlighting research gaps, which necessitated conducting this trial. And the different mode of Action of Trichoderma against pathogenic fungi

Materials and methods:
-this part describes very well by using suitable subheadings. However, it needs a few modifications and details of selecting primers and amplification conditions in the revised version to enhance clarity.

-          The methods of DNA and RNA extraction

-          The ITS and TEF-1α primers sequence as supplemented data

-          The fermentation broth contents: need to be measured using GC-Mas

-          Efficacy of T. koningiopsis C5-9 fermentation controlling B. cinerea in cucumber need positive control as commercial antifungal to good comparison

-          3,5-dinitrosalicylic acid (DNS) 146 method need details

-          gene expression in the RT-qPCR assay were subjected to a one-way analysis of variance 178 (ANOVA). It needs AMOVA also for RNA sequences analysis  

Results and Discussion
-Both parts need to combine and it needs major revision and it needs some figs of gel electrophoresis as supplemented material

-          In order to isolate Trichoderma strains antagonistic to pathogenic fungus B. cinerea 184 under low-temperature conditions, a total of eight Trichoderma strains were isolated and analyzed by in vitro dual-culture assay at 16 °C: what is the period of dual culture

-           Morphological and molecular identification of Trichoderma C5-9: resolution of Trichoderma spores not clear

-          Phylogenetic analysis based on DNA sequence was further performed to confirm the identity of this strain. Based on the ITS sequence and TEF-1α gene: add the accession number of each reference strain in the figures

-          Antifungal activity of the fermentation broth from the T. koningiopsis C5-9: where the antifungal activity data of six Trichoderma strains

-          the fermentation broth produced by Trichoderma C5-9: what is the chemical components of the fermentation broth

Conclusion:
-Improve this part with respect to formulated objectives.
-the 1st para. are not clear.
T. koningiopsis C5-9 directly inhibited B. cinere

References:
-Cross-check the references in the text and reference cite. Few references are not as per journal style in the text as well reference section

Reviewer 2 Report

Transcriptome Analysis Deciphers Trichoderma koningiopsis C5-9 strategies Against Plant Pathogen Botrytis cinerea

“In addition, some Trichoderma biocontrol strains could interact 40 intimately with plant roots”. Name few.

What is the difference between sp. and spp.?

In introduction authors may cite

“Antioxidant and antimicrobial activity displayed by a fungal endophyte Alternaria alternata isolated from Picrorhiza kurroa from Garhwal Himalayas, India,

Biocatalysis and Agricultural Biotechnology, 101955”

What is this “there are still some stain-specific mechanisms”

Why deactivated B. cinerea mycelia was used?

Why authors state this “novel Trichoderma strain C5-9”. How is it considered novel?

How authors selected genes for qRTPCR?

“Early recognition of the host is critical for successful Trichoderma actions”. For this statement authors may use some related work

In discussion, following articles must be cited

Multitrait Pseudomonas sp. isolated from the rhizosphere of Bergenia ciliata acts as a growth-promoting bioinoculant for plants ......................................Front. Sustain. Food Syst. 7 (1097587) 2023

Thermotolerant and halotolerant Streptomyces sp. isolated from Ajuga parviflora having biocontrol activity against Pseudomonas syringae and Xanthomonas campestris .............................Physiological and Molecular Plant Pathology, 102059

Why authors have written few lines for conclusion. Please add more

Minor editing of English language required

Reviewer 3 Report

In my opinion, it is an interesting and well-written work. Certainly, the manuscript will be of interest to readers. I have no comments on the work and I believe that the manuscript can be accepted for publication in its present form.